# The Impact of Mouth-Taping in Mouth-Breathers with Mild Obstructive Sleep Apnea: A Preliminary Study

**DOI:** 10.3390/healthcare10091755

**Published:** 2022-09-13

**Authors:** Yi-Chieh Lee, Chun-Ting Lu, Wen-Nuan Cheng, Hsueh-Yu Li

**Affiliations:** 1Department of Otolaryngology Head & Neck Surgery, Chang Gung Memorial Hospital at Linkou, 5 Fushing St., Taoyuan 333, Taiwan; 2London School of Hygiene and Tropical Medicine, London WC1E 7HT, UK; 3Department of Otolaryngology-Head and Neck Surgery, New Taipei Municipal Tucheng Hospital (Built and Operated by Chang Gung Medical Foundation), New Taipei City 236, Taiwan; 4Department of Sports Sciences, University of Taipei, Taipei 100, Taiwan

**Keywords:** obstructive sleep apnea, snoring, mouth-taping, mouth-breathing

## Abstract

Background: Many patients with obstructive sleep apnea (OSA) are mouth-breathers. Mouth-breathing not only narrows the upper airway, consequently worsening the severity of OSA, but also it affects compliance with nasal continuous positive airway pressure (CPAP) treatment. This study aimed to investigate changes in OSA by the use of mouth tape in mouth-breathers with mild OSA. Method: Mouth-breathers with mild OSA who met inclusion criteria and tolerated the sealing of the mouth were enrolled in the study. We used 3M silicone hypoallergenic tape was used to seal the mouths of the participants during sleep. The home sleep test (HST) used in this study was ApneaLink^®^. Subjects received both a baseline HST and an outcome HST to be used 1 week later while their mouths were taped. The changes between the baseline and the outcome HSTs were compared, and the factors that influenced the differences in the sleep-test parameters after the shift of the breathing route were analyzed. A “responder” was defined as a patient who experienced a reduction from the baseline snoring index of at least 50% under mouth-taping in the HST; otherwise, patients were considered as having a poor response. Results: A total of 20 patients with mild OSA were included. Following the taping of the mouth, a good response was found in 13 patients (65%). The median apnea/hypopnea index (AHI) decreased significantly, from 8.3 to 4.7 event/h (by 47%, *p* = 0.0002), especially in supine AHI (9.4 vs. 5.5 event/h, *p* = 0.0001). The median snoring index (SI) was also improved (by 47%, 303.8 vs. 121.1 event/h, *p* = 0.0002). Despite no significant difference in the mean saturation, improvements in the oxygen desaturation index (8.7 vs. 5.8, *p* = 0.0003) and the lowest saturation (82.5% vs. 87%, *p* = 0.049) were noted. The change in AHI was associated with baseline AHI (r = −0.52, *p* = 0.02), oxygen desaturation index (ODI) (r = −0.54, *p* = 0.01), and SI (r = −0.47, *p* = 0.04). The change in SI was strongly associated with baseline SI (r = −0.77, *p* = 0.001). Conclusions: Mouth-taping during sleep improved snoring and the severity of sleep apnea in mouth-breathers with mild OSA, with AHI and SI being reduced by about half. The higher the level of baseline AHI and SI, the greater the improvement was shown after mouth-taping. Mouth-taping could be an alternative treatment in patients with mild OSA before turning to CPAP therapy or surgical intervention.

## 1. Introduction

Preferentially, people breathe through the nasal route for its physiological functions—heating, humidifying, and filtration—during the daytime and while sleeping. During the daytime, the resistance of the upper airway is similar in nasal-breathing and mouth-breathing [1,2]. During sleep, the resistance is lower in those breathing through the nasal route than in those breathing through the mouth [1,2]. Therefore, the fraction of mouth-breathing normally decreases during sleep. However, for patients with obstructive sleep apnea (OSA), open-mouth-breathing (OMB) is a common symptom during sleep. Mouth-breathing during sleep decreases the retropalatal and retroglossal areas via the posterior displacement of the soft palate and the inferior movement of the mandible, causing a reduction in the length of the upper-airway dilator muscles, which altogether aggravates the severity of OSA [3,4,5,6,7].

Continuous positive airway pressure (CPAP) is the treatment of choice for obstructive sleep apnea (OSA). However, poor adherence to CPAP therapy remains the major issue for OSA patients as nearly one-third of patients cannot tolerate CPAP use after 1 month [8,9]. Studies have shown that mouth-breathing is one reason for poor adherence [10]. Overall, OMB not only affects the severity of OSA [1,5] but also the compliance with nasal CPAP therapy [1,8,10].

Chronic nasal obstruction may lead to mouth-breathing due to decreased nasal flow. Although nasal obstruction is not alone in causing OSA [4], it is frequently associated with mouth-breathing during sleep. Therefore, treating nasal congestion consequently improves mouth-breathing, sleep quality, and compliance with nasal CPAP therapy [11]. The use of mouth tape is one way to maintain nasal-breathing during sleep. A previous pilot study has demonstrated the positive effect of the use of mouth tape against mild OSA (AHI 5, <15) by reducing the AHI and snoring index [12]. However, the previous study used a specially designed tape that limited its use because of low accessibility. In this study, we adopted a widely used 3M tape with advantages (easy to adhere, easy to remove, and non-allergenic) through a simple paste method to seal off the mouth. This tape made mouth-taping easy to perform, and the results are prone to be reproducible for scientific verification. This method can also be implemented easily as a first line of treatment before turning to mainstream therapies, such as CPAP, oral appliances, or surgery. The novelty of this study is that it is a proof of concept and technique—mouth-taping can improve snoring and sleep apnea in mouth-breathers with mild OSA.

The study aimed to investigate the impact of using 3M tape to seal the mouth in mouth-breathers with mild OSA and to explore the factors that influence the reduction of snoring during the shift of the breathing route.

## 2. Methods

### 2.1. Ethics Statement

The study was approved by the Institutional Review Board (IRB) of Chang Gung Memorial Hospital (IRB no. 202201088B0) accompanied by with waivers of the participants’ consent. Linkou Chang Gung Memorial Hospital is the main branch of Chang Gung Memorial Hospital. The IRB of Chang Gung Memorial Foundation is the representative and is responsible for all branches of the Chang Gung Memorial Hospital in IRB review affairs.

### 2.2. Study Population

This retrospective study was conducted between 2020 and 2021 in Chang Gung Memorial Hospital, Linkou Medical Center, Taoyuan, Taiwan. Eligible candidates were diagnosed with adult OSA with a major complaint of snoring.

### 2.3. Study Design

The **inclusion criteria** included patients between the ages of 20 and 60 years, with a body mass index (BMI) of <30 kg/m^2^, AHI < 15 event/h, symptoms of sleep-disordered breathing, witnessed mouth-breathing during sleep, and dryness of throat upon waking in the morning. The **exclusion criteria** were: significant retrognathia; an allergy to mouth tape; intolerance to the sealing of the mouth (the mouth tape was dislodged from the origin site by morning); comorbidity of severe medical diseases; hypertrophy of the palatine tonsil (grade III/IV); previous nose, palate, or tongue surgery; and shift workers. For patients with nasal obstruction, medication and nasal spray were used to ameliorate clinical symptoms and facilitate mouth-taping. Since all subjects visited our clinic for the improvement of snoring, the response was therefore based on the change in snoring index. A **“responder”** was defined as a patient who experienced a reduction from the baseline snoring index of at least 50% under mouth tape in a home sleep test; otherwise, patients were considered as having a poor response.

### 2.4. Home Sleep Test (HST)

Because of the COVID-19 pandemic, we decided to shift away from in-lab sleep tests to HST in some of our sleep-disordered-breathing patients out of a concern for safety and to avoid a delayed diagnosis of OSA. For those patients who had a comorbidity with insomnia, other sleep disorders, or major medical diseases, standard polysomnography was arranged as usual. The HST used in this study was ApneaLink^®^ (ResMed, Sydney, Australia), which is an ambulatory sleep monitor that can detect OSA with acceptable reliability [13,14]. Parameters were measured and analyzed automatically, and then they were reviewed and rescored by an experienced sleep specialist. Reports with incomplete data, such as insufficient recording time (<4 h), a signal that was too weak/sensor loss, or a chaotic signal were excluded, and those patients received new home sleep tests. Subjects received both a baseline HST and an outcome HST to be administered 1 week later under mouth-taping.

### 2.5. Mouth-Taping

The mouth tape used in the study was 3M silicone hypoallergenic tape (1 inch in length). The tape was trimmed to measure 4 cm in length and placed on the philtrum, spanning the upper and lower lips, so as to seal the mouth during sleep. The nocturnal change in airflow during mouth-breathing vs. nasal-breathing is demonstrated in Figure 1.

### 2.6. Data Collection

Basic demographic information (age, sex, and body mass index) was obtained. The variables of the HST included the apnea/hypopnea index (AHI, the sum of AI and HI per hour); the apnea index (AI, decrease in airflow by 90% of baseline for at least 10 s); the hypopnea index (HI, decrease in airflow by 30% to 90% of baseline in addition to a ≥3% reduction of oxygen saturation for at least 10 s per hour) [13]; the supine AHI (AHI of patients in a supine position); the non-supine AHI (AHI of patients in a non-supine position); the mean saturation (the mean saturation during sleep); the lowest saturation (the lowest saturation during sleep) [15]; the 90% saturation percentage (the percentage of total time with an oxygen saturation level lower than 90%); the oxygen desaturation index (ODI, the average number of desaturation events per hour) [16]; and the snoring index (SI, the number of snoring events per hour).

### 2.7. Statistical Analysis

Statistical analysis was conducted using STATA v. 15 (StataCorp LLC, College Station, TX, USA). Data are presented as the median with the interquartile range (IQR, 25th–75th percentile) or number (percentage, %). Wilcoxon signed-rank tests were used to compare the baseline and outcome HST data, and Chi-square tests were used for comparing the categorical data between the groups. Spearman rank-order correlation analysis was used to evaluate the relationships between variables. A *p*-value < 0.05 was regarded as statistically significant.

## 3. Results

A total of 20 patients met the inclusion criteria and were included for further analysis. Table 1 summarizes the distribution of the baseline characteristics. The study population consisted of 19 men (95.0%) and 1 woman (5.0%) with a median age of 38 years (IQR: 30–43 years old). The median BMI was 24.5 kg/m^2^ (IQR: 23.6–26.0 kg/m^2^). Median AHI values before the use of the mouth tape were 8.3 (IQR: 6.2–12.9) events per hour.

A comparison of the HST data from before and after the use of the mouth tape is shown in Table 2. The severity of OSA decreased in all participants after the mouth tape was used. Specifically, a significant reduction of values was seen in the AHI (*p* value = 0.0002), the AI (*p* value = 0.002), and the HI (0.003) (see Figure 2). The individual SI improved significantly after mouth-taping (see Figure 3). Moreover, most of the participants experienced worse symptoms in the supine position (median AHI: 9.4, IQR: 7.3–15.7) than in the non-supine position (median AHI: 3.2, IQR: 0.1–5.0). Mouth-tape use predominantly improved the OSA symptoms in the supine position, with a reduction of the median AHI from 9.4 to 5.5 events per hour (*p* value = 0.0001).

An association between the use of mouth tape and a change in the mean saturation and 90% saturation during sleep was not found in the study. However, significant improvements in ODI (*p* value = 0.0003) and SI (*p* value = 0.0002) were shown, from 8.7 to 5.8 and 303.8 to 121.1, respectively. No significant factors (including age, sex, BMI, and all sleep parameters) affecting response (>50% reduction from baseline snoring index) of the use of mouth tape were found in the analysis. The correlations between variables contributing to the change in the AHI and SI were also examined. A statistically significant correlation of change in AHI was seen in the baseline AHI (r = −0.52, *p* value =0.02), the baseline oxygen desaturation index (ODI) (r = −0.54, *p* value =0.01), and the baseline SI (ρ = −0.47, *p* value = 0.04). Furthermore, a change in SI was associated with a baseline SI (r = −0.77, *p* value =0.001) (see Figure 4).

## 4. Discussion

Studies have shown an association between OMB and OSA [1,17]. OMB elongates and narrows the upper airway, which negatively affects the severity of OSA [6]. Therefore, a higher percentage of mouth-breathers is found among people with OSA. Humans preferentially breathe through the nasal route during the daytime and while sleeping for the benefit of physiological functions. However, during sleep, people might have OMB resulting from chronic nasal obstruction or habitual mouth-breathing [3]. For people with nasal obstruction, medication and nasal spray were given before the use of mouth tape. In the study, we evaluated the potential benefits of using mouth tape in treating mild OSA patients with open-mouth-breathing (OMB). The study only enrolled patients who tolerated the use of mouth tape during sleep. The hypothesis is that taping the mouth in these patients during sleep may improve their OSA by causing them to switch from oral- to nasal-breathing.

The results showed improvements in most of the parameters in the sleep tests. There was a significant decrease (*p* = 0.0002) in AHI after the use of mouth tape. The level of the AHI was even lowered to the normal range in some participants.

In the study, most (75%) of the participants had positional sleep apnea [18,19]. The supine AHI experienced a significant reduction after the use of mouth tape (p-value: 0.0001). In a previous study, positional sleep apnea was demonstrated as being more common in mild OSA patients [19], and the obstruction site was more likely in the soft palate [20]. Lee et al. also demonstrated significant changes in the retropalatal space between the closed-mouth and the open-mouth positions due to the posterior displacement of the soft palate when the mouth is open [1]. More-elongated and narrower upper-airway spaces were also found in patients with OSA using 3D multidetector computed tomography (3D MDCT) [6]. We can infer that, after the treatment of any nasal obstruction and the switch to nasal-breathing via the taping of the mouth, the retropalatal space might be widened and therefore improve snoring and the severity of OSA. Since ODI and AHI have a good concordance, a significant reduction of ODI after mouth-taping is also conceivable [21].

No significant factor affecting the effectiveness of the use of mouth tape was seen in the analysis. However, in positional sleep apnea patients (*n* = 15), 11 (73.3%) of them were classified as responders. In comparison, only 2 (40%) out of the 5 non-positional sleep apnea patients were responders. Although the results did not show a statistical significance, that could be attributable to the small sample size as the percentage difference (33.3%) was substantial. A study with a larger sample size could be performed to investigate the effectiveness.

A statistically significant negative correlation was observed between the change of the AHI with the baseline AHI, ODI, and SI. For patients with a higher level of baseline AHI, ODI, and SI, the effectiveness of mouth-taping was greater. The improvement of the AHI was greater in the more severe cases of mild OSA, but not in the moderate or severe cases of OSA (AHI > 15). Mouth-taping is not recommended for moderate or severe OSA patients because it may impose dangers rather than benefits in these patients. Besides, a strong negative correlation between the change in SI and the baseline SI was found in the analysis, which could be a good predictor of the effectiveness of mouth-taping treatment.

In recent years, oral appliances such as mouth tape and oral shield devices have been invented and have been shown to be effective in clinical use [22]. Foellner et al. [23] showed the positive effect of using an oral shield device concomitantly with nasal CPAP. Huang et al. [12] also demonstrated the promising result of using novel oral patches for treating mild OSA. However, the device and the patch are of limited use due to their accessibility. The 3M mouth tape is already a product on the market, and it is prevalent and accessible. Moreover, the product is affordable and user-friendly. Our study highlights the utility of 3M mouth tape as an easy and inexpensive tool to mitigate the severity of OSA in selective patients.

There are some limitations to this study. First, the study has a small sample size without a control group, and no comparison could be made to determine whether there was a placebo effect. Second, this is a retrospective study with a short follow-up period, and no long-term effect was evaluated. Besides, the home sleep test used in the study may underestimate the severity of OSA. Also, the low proportion of women is another drawback. Finally, the follow-up period is short, and there could possibly be an issue of low adherence to mouth-taping in the long term. For further research, prospective studies with larger sample sizes and control groups can be done to assess the efficacy of the mouth tape.

## 5. Conclusions

Our study provided a simple and effective treatment modality using 3M mouth tape for mild OSA patients with open-mouth-breathing. The AHI and SI were reduced by nearly half after mouth-taping during sleep; the more severe the baseline AHI and SI, the greater the improvement after mouth-taping. For mild OSA mouth-breathers, mouth-taping could be an alternative treatment before CPAP therapy or surgical intervention are tried.

## Figures and Tables

**Figure 1 healthcare-10-01755-f001:**
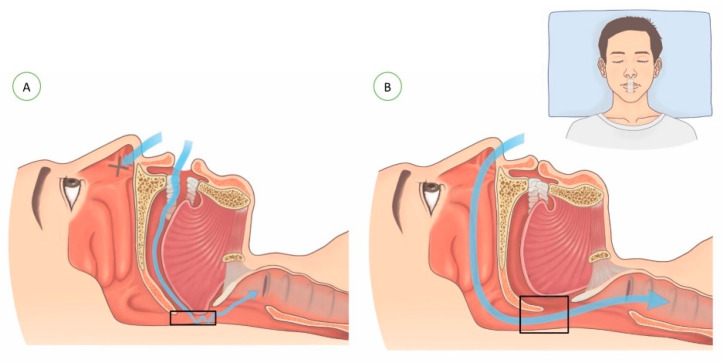
Figures demonstrating the breathing routes of (**A**) mouth-breathing and (**B**) nasal-breathing after mouth-taping.

**Figure 2 healthcare-10-01755-f002:**
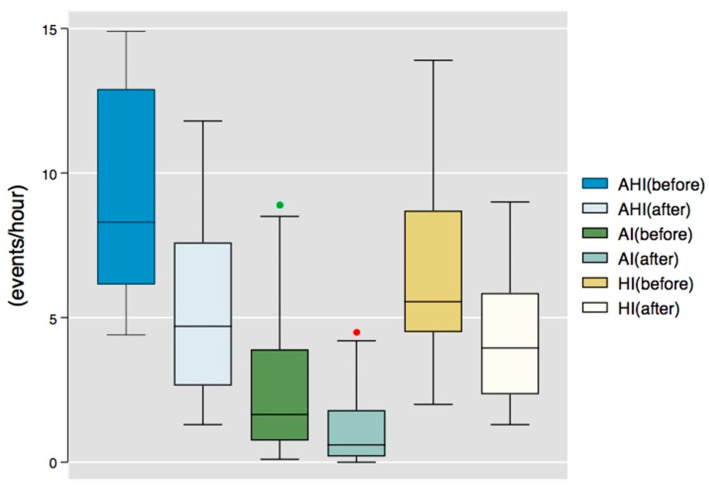
Box graph of apnea/hypopnea index (AHI), apnea index (AI), and hypopnea index (HI) before and after mouth-taping. (The green and red dots represent outliners of each parameter).

**Figure 3 healthcare-10-01755-f003:**
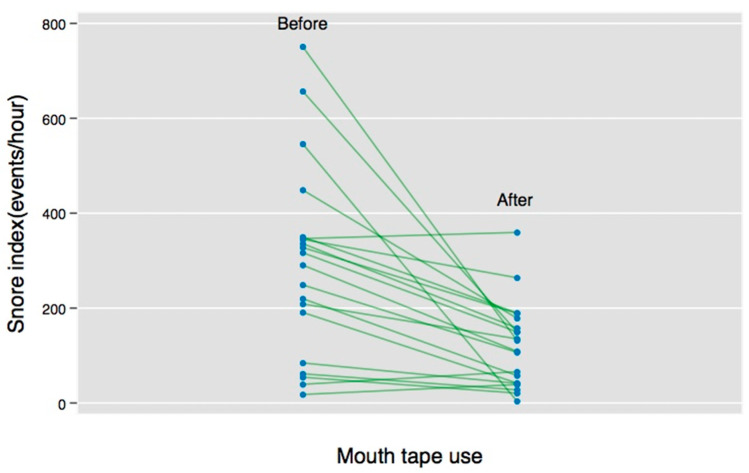
Individual change in snoring index (SI) before and after mouth-taping.

**Figure 4 healthcare-10-01755-f004:**
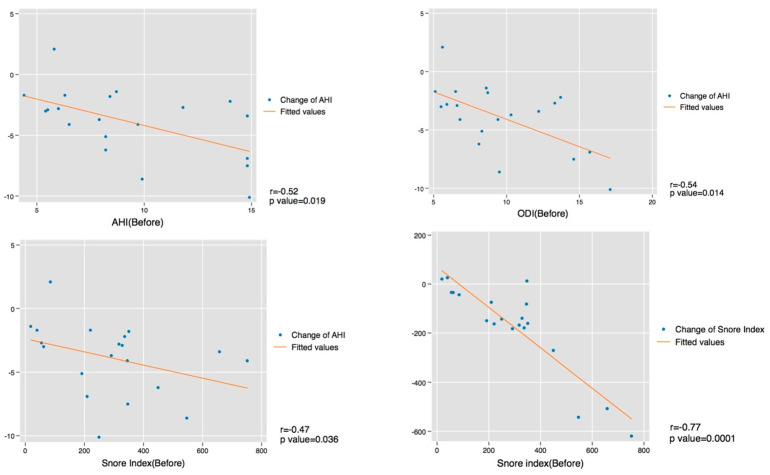
Correlation tests between variables (including the change of AHI from baseline AHI, ODI, and snoring index, and the change in snoring index from baseline snoring index).

**Table 1 healthcare-10-01755-t001:** Demographic information.

	n (%) or Median (IQR)
**Age (in years)**	38 (30–43)
**Sex (male/female)**	19 (95.0%)/1 (5.0%)
**Height (cm)**	175 (170.9–177.25)
**Weight (kg)**	74 (67.7–79.5)
**BMI (kg/m^2^)**	24.5 (23.6–26.0)

**Table 2 healthcare-10-01755-t002:** HST data from before and after mouth-taping.

	Mouth-Breathing		Nasal-Breathing		Changes (Percentage %) †		
HST Parameters	Median	IQR	Median	IQR	Median	IQR	*p*-Value
**AHI (event/h)**	8.3	6.2–12.9	4.7	2.7–7.6	−47%	−59%~−23%	0.0002
**AI (event/h)**	1.7	0.8–3.9	0.6	0.2–1.8	−51%	−76%~−18%	0.002
**HI (event/h)**	5.6	4.5–8.7	4	2.4–5.9	−44%	−56%~−14%	0.003
**Supine AHI** **(event/h)**	9.4	7.3–15.7	5.5	3.8–8.9	−41%	−54%~−41%	0.0001
**Non-supine AHI (event/h)**	3.2	0.1–5.0	0.6	0.1–2.2	−55%	−95%~−44%	0.03
**Mean saturation (%)**	95	94.5–95.5	95	94.5–96	0%	−1%~1%	0.9
**Lowest saturation (%)**	82.5	80.5–88	87	84–89.5	2%	−1%~8%	0.049
**90% saturation (%)**	1	0–2.5	0	0–1	−67%	−100%~0%	0.12
**ODI (event/h)**	8.7	6.6–12.8	5.8	3.3–8.1	−36%	−58%~−20%	0.0003
**Snoring index (event/h)**	303.8	137.8–348.3	121.1	41.8–168.4	−47%	−59%~−23%	0.0002

HST: home sleep test, AHI: apnea/hypopnea index; AI: apnea index; HI: hypopnea index; ODI: oxygen desaturation index; IQR: interquartile range. † changes (percentage %) = (value of nasal-breathing − value of mouth-breathing)/value of mouth-breathing.

## Data Availability

The data is not publicly available due to the regulation of our institution and the protection of patients’ privacy, particularly in the small sample size group. However, the data presented in the study is available on request from the corresponding author for further research.

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
