# Peer review of "The Impact of Mouth-Taping in Mouth-Breathers with Mild Obstructive Sleep Apnea: A Preliminary Study"

_healthcare, 2022, doi:10.3390/healthcare10091755_

Round 1

Reviewer 1 Report

The authors are presenting mouth taping treatment for mild OSA patients before CPAP therapy or surgical interventions.

1. The introduction is well written and clearly mentions the background. However, it may please also include how the current study contributes as compared with the existing studies about mouth taping specifically and highlight what exactly is the main novelty here in that regards.

2. Also, please consider stating the percentage where "the percentage difference was substantial " is mentioned on page 6/8 given that the statistical differences doesn't exist.

3. Please consider including formulae in the manuscript for each metric measurements shown in table 2 such as Snore Index etc for better readability.

4. Please consider increasing the font size for figure axes, legends and labels.

5. Please elaborate the caption for figure 4. What are the variables? What does it mean/imply? It's not readable. 

Overall, the paper can be accepted with the above minor revisions. 

Reviewer 2 Report

I have found the manuscript very intersting and well written. It adds new options for the OSAS management.

A linguistic revision by mother tongue is my only suggestion

Reviewer 3 Report

This manuscript is a retrospective study on mouth taping in mild obstructive sleep apnea mouth breathers.

It is an interesting and well-written paper.

I recommend its publication.

However, it needs minor improvements:

1.- Mat & Method

Sample size calculations are missing.

One week between first home sleep test and the second one is very short.

2.- Results

Mouth taping was efficacious, with a 47% reduction in AHI.

However, how many patients were adherent to taping? How many hours per night?

3.- Discussion

Consider mentioning:

Kim EJ, Choi JH, Kim KW, Kim TH, Lee SH, Lee HM, Shin C, Lee KY, Lee SH. The impacts of open-mouth breathing on upper airway space in obstructive sleep apnea: 3-D MDCT analysis. Eur Arch Otorhinolaryngol. 2011 Apr;268(4):533-9. doi: 10.1007/s00405-010-1397-6.

Jau JY, Kuo TBJ, Li LPH, Chen TY, Lai CT, Huang PH, Yang CCH. Mouth puffing phenomena of patients with obstructive sleep apnea when mouth-taped: device's efficacy confirmed with physical video observation. Sleep Breath. 2022 Mar 11. doi: 10.1007/s11325-022-02588-0.

Limitations should include short follow-up and possible low adherence.
